# Microvascular Alterations of Peripapillary Choriocapillaris in Young Adult High Myopia Detected by Optical Coherence Tomography Angiography

**DOI:** 10.3390/jpm13020289

**Published:** 2023-02-04

**Authors:** Jie Lei, Yuanyuan Fan, Yan Wu, Songtao Yuan, Yurong Ye, Kun Huang, Qiang Chen, Bin Yang, Ping Xie

**Affiliations:** 1Department of Ophthalmology, The First Affiliated Hospital of Nanjing Medical University, Nanjing 210029, China; 2School of Computer Science and Engineering, Nanjing University of Science and Technology, Nanjing 210094, China; 3Department of Ophthalmology, Yangzhong People’s Hospital, Yangzhong 212299, China

**Keywords:** choroidal microvascular dropout, high myopia, optical coherence tomography angiography, peripapillary atrophy

## Abstract

(1) **Background**: The microstructural alterations of the peripapillary choriocapillaris in high myopes remain elusive. Here, we used optical coherence tomography angiography (OCTA) to explore factors involved in these alterations. (2) **Methods**: This cross-sectional control study included 205 young adults’ eyes (95 with high myopia and 110 with mild to moderate myopia). The choroidal vascular network was imaged using OCTA, and the images underwent manual adjustments to determine the peripapillary atrophy (PPA)-β zone and microvascular dropout (MvD). The area of MvD and the PPA-β zone, spherical equivalent (SE), and axial length (AL) were collected and compared across groups. (3) **Results**: The MvD was identified in 195 eyes (95.1%). Highly myopic eyes exhibited a significantly greater area for the PPA-β zone (1.221 ± 0.073 vs. 0.562 ± 0.383 mm^2^, *p* = 0.001) and MvD (0.248 ± 0.191 vs. 0.089 ± 0.082 mm^2^, *p* < 0.001) compared with mildly to moderately myopic eyes, and a lower average density in the choriocapillaris. Linear regression analysis showed that the MvD area correlated with age, SE, AL, and the PPA-β area (all *p* < 0.05). (4) **Conclusions**: This study found that MvDs represent choroidal microvascular alterations in young-adult high myopes, which were correlated with age, SE, AL, and the PPA-β zone. In this disorder, OCTA is important for characterizing the underlying pathophysiological adaptations.

## 1. Introduction

Myopia, the major cause of vision impairment and the second most common cause of blindness [1], affects nearly two billion people (28.3% of the global population), which includes 277 million people (4.0% of the global population) with high myopia [2]. With the increase in refractive error and axial length (AL), the risk of adverse ocular tissue alterations—including posterior staphyloma, retinoschisis, retinal detachment, choroidal/scleral thinning, lacquer crack formation, myopic choroidal neovascularization, and glaucoma [3]–increases dramatically, especially in populations with high levels of myopia. Accordingly, there has been a push to better control the onset and worsening of myopia and to place this public health problem high on the agenda. While improved strategies—such as the use of atropine or anti-hypoxia drugs, wearable optical devices, surgical management, and orthokeratology, for example—have been employed to intervene early in and limit myopia progression, the exact pathological mechanism of the myopia of the human eye remains elusive, hindering the development of successful treatment and the reduction in the burden of disease [4]. Previous studies have suggested that high myopia is associated with decreased blood supply [5,6,7]. Li et al. [5] found that the density of both superficial and deep microvascular plexuses dropped remarkably in highly myopic patients with normal visual ability before irreversible retinal damage. Sung et al. [6,7] also described a lower retinal vessel density and a larger foveal avascular zone in high myopia. However, further evidence of vascular change in the choroidal level is limited.

Not merely regarded as a densely vascularized structure lying between the retina and sclera, the choroid possesses unexpectable elasticity in axial elongation [8] and perhaps accommodates intrinsic neurons [9] and lymphatics [10]. The choriocapillaris, restricted to the innermost part of the choroid, is composed of highly compacted capillaries separated by many tiny inter-capillary spaces that may grow with age and disease [11,12]. Such inter-capillary spaces were strikingly captured by optical coherence tomography angiography (OCTA) and represented in the form of small dark regions, known as flow voids or microvascular dropouts (MvD) [13]. Lee et al. [14] confirmed that the MvDs found by OCTA were coincident with the perfusion defects of indocyanine green angiography in glaucoma. Though there have been studies that focus on the central macular choroidal features in eyes with high myopia [15,16,17], less attention has been paid to the peripapillary alterations by OCTA. This may have resulted from having limited avenues to represent the peripapillary blood state, as the image quality is easily affected by the distortion of the optic disc, a symptom usually seen in high myopes, thus creating unfavorable errors and causing misleading conclusions to be made. MvD, a newly-proposed concept that had never been investigated among high myopic populations may bring improved precision in evaluating choroidal perfusion defects.

OCTA provides the non-invasive, rapid, and reproducible image acquisition of multiple layers of the posterior segment without the need for fluorescent dye injection by using phase or amplitude decorrelation to identify the motion contrast of the blood flow [18,19,20]. It shows a detailed visualization of microvessels and an analysis of blood signals at the micrometer level so that MvDs, subtle sectoral capillary voids ignored by traditional imaging techniques, can be detected and shown in the choroid layer in en face images, usually appearing in the peripapillary atrophy (PPA) zone. PPA is the atrophy and thinning of the local retina and choroid occurring in the juxtapapillary area adjoining the optic disc margin, mainly seen in elderly, glaucomatous, or highly myopic populations. It is classified into two categories based on histological characteristics. The α-zone features irregular hypopigmentation and hyperpigmentation due to irregularities in the parapapillary RPE and the intimated thinning of the chorioretinal tissue layer [21,22]. It is histologically defined by the presence of Bruch’s membrane (BM) and an irregularly arranged RPE, which, in some cases, looks like it is being rolled up on its end [23]. The β-zone features the disappearance of the retinal pigment epithelium (RPE) and the obvious atrophy of the choriocapillaris, with good visibility of the large choroidal vessels and the sclera [24]. It is histologically defined as the presence of BM and the absence of the RPE [25] (Figure 1B,C). If both zones are present, the α-zone is always more peripheral than the β-zone and represents less severe structural damage. Previous studies on glaucoma have confirmed that MvD is more frequently present in eyes with a β-zone than those without a β-zone in the PPA area [26,27].

In the present study, we employed OCTA to evaluate the relationship among PPA, MvD, and high myopia, and to investigate the underlying pathophysiological mechanism of the disorder which may facilitate the early management of high myopia.

## 2. Materials and Methods

The subjects were selected from populations at the First Affiliated Hospital of Nanjing Medical University (Nanjing, China) from April 2019 through January 2020. Written informed consent to participate was obtained from each of the enrolled subjects. The study protocol was registered with clinicaltrial.gov (accessed on 27 December 2022, code number NCT04255524) and approved by the Institutional Review Board of First Affiliated Hospital of Nanjing Medical University, and it followed the tenets of the Declaration of Helsinki.

### 2.1. Study Subjects

Subjects were divided into two groups based on the spherical equivalent (SE), which was calculated as the spherical dioptric power plus one half of the cylindrical dioptric power. Subjects with a SE between −10.0 diopters (D) and −6.0 D were allocated to the high myopia group, while subjects with a SE between −6.0 D and −0.5 D were in the mild to moderate myopia group.

Each subject underwent comprehensive ophthalmic examinations, including slit-lamp biomicroscopy using the Haag-Streit BM 900 slit-lamp microscope (Haag-Streit, Köniz, Switzerland), intraocular pressure using the Canon TX-20 non-contact tonometer (Canon, Tokyo, Japan), best-corrected visual acuity, a refraction test using autorefractometer RC-5000 (Tomey, Nagoya, Japan) and phoropter RT-5100 (Nidek, Tokyo, Japan), AL measurement using Nidek-AL Scan optical biometry (Nidek, Gamagori, Japan), fundus color photography and red-free fundus photography using the Canon CR-2 Plus AF non-mydriatic retinal camera (Canon, Tokyo, Japan), and OCTA using Optovue Angio VueTM System (Optovue, Fremont, CA, USA). All the examinations were completed by one professionally trained ophthalmologist (J.L).

The main inclusion criteria were subjects who were: (1) between 18 and 40 years old, with (2) a best-corrected visual acuity of ≥20/20 and a SE ranging from −0.5 to −10.0 D, (3) an intraocular pressure of <21 mmHg and <5 mmHg of difference between the two eyes, (4) no history of elevated intraocular pressure, and (5) no glaucomatous appearance. The absence of a glaucomatous appearance was defined as follows: a cup-disc ratio of <0.5 with ≤0.2 of difference between the two eyes, an intact neuroretinal rim, no disc hemorrhages, notches or a localized pallor, and a normal retinal nerve fiber layer thickness without any defects. The exclusion criteria were subjects with: (1) a reluctance to sign informed consent or receive consecutive following-up; (2) an incapability of keeping eyeballs stable, undergoing ocular examinations, or producing good-quality images; (3) an astigmatism beyond ±4.0 D; (4) any history of intraocular or refractive surgery; and (5) the presence of any retinal or neurological diseases (other than myopic degeneration), opaque media, or glaucoma.

### 2.2. OCTA and Determination of the Presence of a MvD

The optic nerve and peripapillary area were imaged using a commercially available spectral-domain Optovue Angio VueTM OCTA system (Optovue, Fremont, CA, USA) operating at a central wavelength of 840 nm, an acquisition speed of 70,000 A-scans per second, and axial and transverse resolutions of 5 and 15 μm in tissue, respectively. Scans were obtained from 4.5 × 4.5 mm cubes, with each cube consisting of 400 clusters of twice-repeated B-scans centered on the optic disc. Automatic layer segmentation in the “angio-disc” pattern, performed by the built-in software controlling the OCT instrument, generated scanning laser ophthalmoscope images of the disc and peripapillary region and en face projections to visualize the vasculature and structure of the vitreous/retina layer from the vitreous body to the outer plexiform layer (50 μm below internal limiting membrane), the retinal posterior capillary layer from the internal limiting membrane to nerve fiber layer, and the choroid layer below the RPE.

All the OCTA images were exported in PNG format (Figure 1A). The color fundus image, scanning laser ophthalmoscope image, and OCTA image of each participant were carefully evaluated by two ophthalmologists (J.L and Y.Y.) to contour the PPA-α and PPA-β zone (Figure 1B,C) under the supervision of two veteran ophthalmologists (P.X and B.Y.). On the OCTA images, we manually defined “pixel” as the minimum unit of area measurement, that is, a pixel was equal to a cluster of a 4.5 × 4.5 mm cube (126.5625 μm^2^). When assessing MvD, the dataset of OCTA images was then transferred to Nanjing University of Science and Technology (Nanjing, China) to be analyzed (K.H. and Q.C.). The appropriate threshold was identified as regions with a grey value less than 50 and an area exceeding 20 pixels (Figure 1A). The area of MvD and the PPA-β zone was calculated by PyCharm software counting the number of pixels that met this standard. Large vessels were removed from the OCTA image in advance, without whose influence the mean density of the choroidal blood flow could be more accurately calculated as the area occupied by blood signals in the scanned region. All of the included OCT B-scan images had scan quality scores of ≥6/10. Any image with a double-vessel pattern, motion artifacts, or segmentation errors extending more than three lines was excluded from the analysis.

### 2.3. Data Analysis

All data were expressed as the mean ± standard deviations and were analyzed statistically using SPSS software (version 25.0; SPSS, Chicago, IL, USA). Demographic, ocular, and systemic characteristics were compared between groups using the independent samples’ *t*-test for numerical variables. Correlations between the area of the PPA-β zone and relative factors were determined using the Spearman correlation test. The criterion for statistical significance was *p* < 0.05.

## 3. Results

This study initially included 220 eyes from 110 subjects with myopia, of which 15 eyes were excluded due to poor OCTA image quality. A total of 95 eyes were in the high myopia group, and 110 eyes were in the mild to moderate myopia group. MvD was identified in 195 of the remaining 205 eyes (95.1%) in the PPA-β zone.

Demographics of the enrolled subjects are shown in Table 1. There was no significant difference in sex (*p* = 0.873) or age (*p* = 0.517) between the high myopia (*n* = 95) and the mild to moderate myopia group (*n* = 110). The mean SE and AL were −7.14 ± 0.927 D and 26.65 ± 0.88 mm in the high myopia group, and −2.51 ± 2.10 D and 24.36 ± 1.15 mm in the other, respectively. Both parameters were significantly different between the two groups (*p* < 0.05). 

In the high myopia group, the mean area of the PPA-β zone was significantly greater than that of the mild to moderate myopes (1.221 ± 0.073 vs. 0.562 ± 0.383 mm^2^, *p* < 0.05, Figure 2A). A significantly larger area of MvD was also observed in highly myopic eyes (0.248 ± 0.191 vs. 0.089 ± 0.082 mm^2^, *p* < 0.05, Figure 2B). Meanwhile, the high myopia group exhibited a lower mean density of choroidal blood flow (157.60 ± 218.42 vs. 404.88 ± 432.47) in comparison to the mild to moderate myopia group, which marked a significant difference between the two groups (*p* < 0.05, Figure 2C).

MvD was universally identified within the β-zone and additionally involved the α-zone in eyes with both β- and α-zones. As is shown in Table 2, 22.1% of subjects in the mild to moderate myopia group were found to have MvDs in the PPA-α area, while this figure was 57.3% in the high myopia group. While the area of the PPA-α appeared larger in the high myopia group, no significant difference was detected between groups in either the area of MvD in the PPA-α zone (*p* = 0.210) or the mean density of the choroidal flow in the PPA-α zone (*p* = 0.159).

A linear regression analysis showed that age (*p* < 0.05), SE (*r* = −0.484, *p* < 0.05, Figure 3A), AL (*r* = 0.477, *p* < 0.05, Figure 3B), and the area of the PPA-β zone (*r* = 0.894, *p* < 0.05, Figure 3C) were all independent and significant factors that affected the MvD area in myopic degeneration. 

## 4. Discussion

In this study, OCTA helped us to reveal the microcirculatory alterations of the juxtapapillary area at the choroid level in young adults with high myopia. The gold standard diagnosing modality for retinal and choroidal vasculature has been fluorescein angiography, since 1961, for its revolutionized ability to reflect patterns of dye transit and leakage dynamically [28]. The introduction of indocyanine green, wide-field image acquisition, and confocal scanning laser ophthalmoscopy further expanded its application profiles and possibilities. Nevertheless, a major problem is that an entire choroidal capillary network and a direct visualization of nascent vessels are not possible to be imaged by traditional angiography. OCTA boasts high-speed OCT scanning to sense blood flow by analyzing signal decorrelation between scans. In contrast, in stationary areas, the movement of erythrocytes within vessels produces decorrelated signals, allowing for a direct visualization of both large vessels and minor circulatory changes from the internal limiting membrane to the choroid. Additionally, without the need for exogenous fluorescent dye injection or the risk of allergic symptoms, OCTA harbors the ability to image three-dimensional choroidal vasculature detailly. This powerful feature is conducive to acquiring perfusion data in the absence of obvious morphological changes [18]. In this study, we used a spectral-domain OCT system to evaluate blood flow status more precisely than the traditional time-domain OCT system allows. Recently, the emergence of the newest generation of the swept-source OCT (SS-OCT) with a longer wavelength (1050~1060 nm) has made it possible for the light to penetrate the RPE and to image the choroidal vasculature more clearly. It is apt to adopt SS-OCT in future studies for a higher imaging speed and a better signal-to-noise ratio.

Our results showed that the high myopia group exhibited a lower density of choroidal blood flow and a larger PPA-β area (Figure 2A,B), indicating that myopia progression promotes vascular and structural perturbations. These findings are in agreement with previous studies that demonstrated less choroidal blood perfusion in high myopia using pulse amplitude and other relevant parameters [29,30]. In parallel, a decreased retinal nerve fiber layer and choroidal thickness accompanied by subsequent disc hemorrhage have been linked to myopia [31]. However, whether structural or vascular alterations change first, or whether both change simultaneously, has troubled experts and researchers for a long time. On the one hand, a decline in choriocapillary perfusion may arise from reduced blood demands, secondary to choroidal degeneration occurring in highly myopic eyes. Jing Zhao et al. [32] found that axial-elongation-associated choroidal thinning affected Haller’s and Sattler’s layers more markedly than it affected the small-vessel layer. A lack of blood supply from the medium- and large-sized choroidal vessels in Haller’s and Sattler’s layers may also contribute to lower choriocapillary perfusion. Additionally, it is also speculated that the progression of optic disc tilt during global elongation may directly and mechanically deteriorate the peripapillary microvasculature [33]. On the another hand, hypoxia has been regarded as essential for scleral extracellular remodeling during myopia development [34]. Visual perception may induce cell-signaling-pathway cascades that affect choroidal blood perfusion and scleral oxygenation, triggering a series of downstream events that diminish extracellular integrity and exacerbate scleral extension. This is a chicken-and-egg question in light of which comes first and which warrants further prospective studies to determine the sequence between structural and vascular alterations.

With the development of OCTA, choroidal MvD has frequently been reported in glaucoma cases as an indicator of vascular changes [35,36]. Numerous studies have proved its relation to structural and functional alterations [37,38,39,40,41]. What is well known is that glaucomatous eyes exhibit a higher prevalence of myopia and that myopia is one of the risk factors that induce primary open-angle glaucoma [42], based on which we transplanted the concept of MvD in the context of myopia and observed that 95.1% of the surveyed eyes had MvD, confirming the existence of dropouts in myopia for the first time. Although the MvD area in the mild to moderate myopia group was far lower than that in the high myopia group (Figure 2C), the appearance of dropouts in early myopia suggested that minor vascular defects may occur as soon as myopia develops. Alternatively, MvD detected in both glaucomatous and myopic eyes also emphasized their potential connections in etiology and encouraged more efforts into preventing one from experiencing the detrimental effects of another.

Notably, the association between the microstructure of the PPA-β zone and retinal vessel density has been investigated by Sung et al., [7]. The width of the PPA zone was negatively correlated with a superficial and deep parapapillary vessel density. At the choroidal level, Hu et al., [43] acquired similar outcomes. Our results, from another perspective of MvD, confirmed the association between choroidal circulation and retinal structure. MvD was more frequently detected in the PPA-β zone and more relevant to the PPA-β area, age, and AL, confirming that the drop of choroidal perfusion is linked to retinal structural deterioration and myopia progression. The choroid, situated between the sclera and BM, is the destination of 70% of all the ocular blood flow for its highly compacted capillary network [44]. Lying in the outer retina, photoreceptors are provided oxygen and nourishment by choroidal circulation [45], which have the highest rate of oxygen use per unit weight of tissue in the body [46], to sustain normal morphology and function. It makes sense that during the elongation of progressively myopic eyes, the choroid is stretched and the vessels are impaired or even destroyed. Abnormalities in the retina state may be induced following the reduction in the choroidal blood supply.

PPA-α is histologically defined by the presence of BM and an irregularly arranged RPE, which in some eyes appears to be rolled up on its end, while PPA-β is characterized by the presence of BM and the absence of an RPE [23,25]. Corresponding to the histological anatomy, the PPA-β zone represents an absolute scotoma in perimetry and the PPA-α zone represents a relative scotoma [47,48,49]. Therefore, PPA-β is the more severe form of retinal structural destruction developed from PPA-α. Here, we found that MvDs could be detected in almost all PPA-β zones but also in some PPA-α zones, which indicated the presence of microcirculatory defects in the choroid before the retinal degeneration from PPA-α into PPA-β. In other words, the changes in blood flow at the choroid level may occur prior to RPE and choroidal atrophy. Such observations are explainable by the insufficiency of life-sustaining materials originating from choroidal ischemia, the latter being an integral part of the apoptosis of RPE cells and consequent PPA development. It also should be noted that the present study has several limitations. First, because of the cross-sectional property, this study was conducted at a specific time point instead of within a period, which means we could not determine the cause and effect between MvD and the PPA-β. Second, we did not represent the choroidal vascular network during different stages of high myopia due to our relatively unsatisfactory sample size and the lack of pathologically myopic populations. Furthermore, our expectations for OCTA must be tempered given the challenges, such as a failure to reflect patterns of ischemia, occlusion, and leakage in the periphery; unavoidable projection artifacts caused by the moving shadows of blood cells; and the lack of a standardized automated segmentation algorithm [18]. Biases may also be produced by optical magnification for severe myopia due to the influence of axial length [50].

In conclusions, we characterized the peripapillary alterations of the microcirculatory network at the choroid level in highly myopic eyes with the assistance of OCTA. MvD, more frequently appearing in high myopes than in mild to moderate ones, correlated with age, AL, SE, and the area of the PPA-β zone, which may play an important role in the early detection and diagnosis of myopia-associated retinopathies. Collectively, quantitative analysis from OCTA images of peripapillary microvasculature is extraordinarily conducive to digging out the underlying pathological mechanisms behind myopia. It is imperative to run further longitudinal investigations to elucidate if these mechanical and anatomical changes occur before or after the vascular changes.

## Figures and Tables

**Figure 1 jpm-13-00289-f001:**
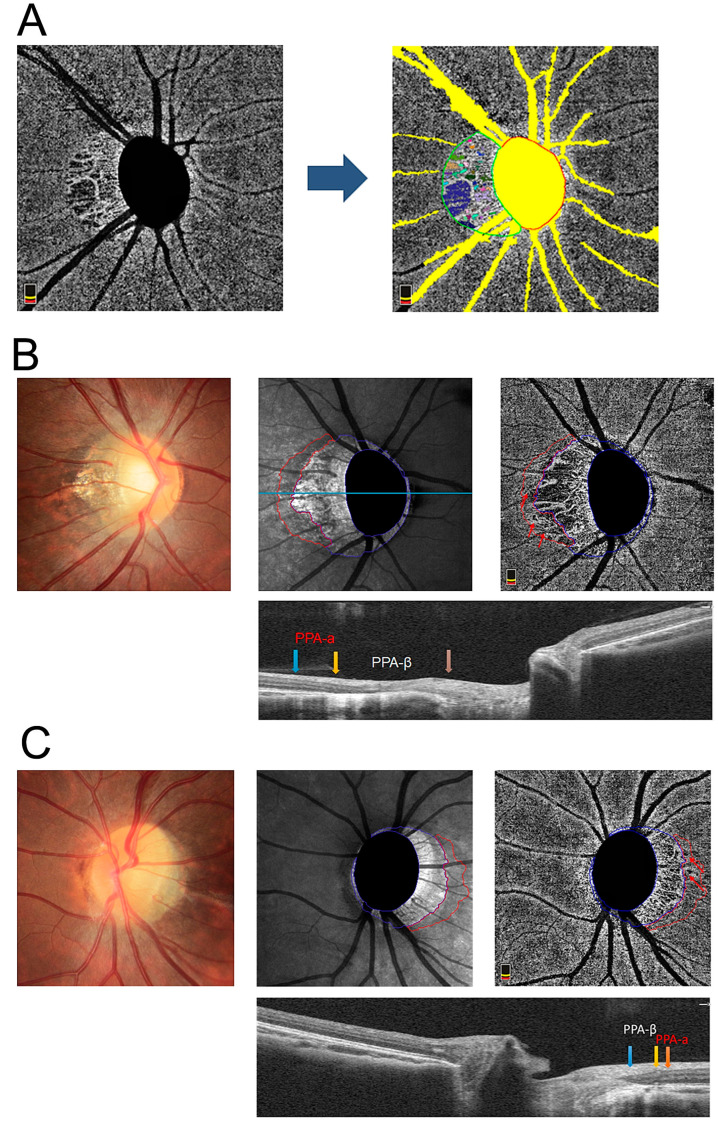
Case of high myopia with peripapillary microvascular dropout (MvD) in both peripapillary atrophy (PPA)-β and PPA-α zone. (**A**) Scanning laser ophthalmoscope image (SLO) of OCTA was proceeded with auto-segment of MvD (color areas in green-circled region) after removinglarge vessels and. (**B**,**C**) Peripapillary multimodal images of a 23-year-old female with −6.00 diopters (D) and a 21-year-old female with −6.50 D. The PPA-α zone is circled blue on SLO image and correspondingly lies between blue arrow and yellow arrow on OCT image. The PPA-β zone is circled red on SLO image and correspondingly lies between the yellow arrow and orange arrow on the OCT image. Red arrow indicates the MvD within the PPA-α zone.

**Figure 2 jpm-13-00289-f002:**
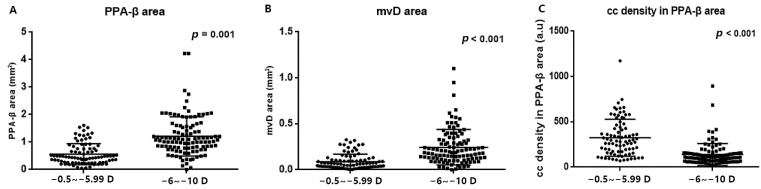
Comparison of the area of atrophy and blood flow density in choroid between high myopia group and mild to moderate myopia group. The area of (**A**) PPA-β zone and (**B**) MvD were both larger in high myopia group (both *p* < 0.05), but (**C**) the mean blood flow density was significantly lower in high myopia group (*p* < 0.05).

**Figure 3 jpm-13-00289-f003:**
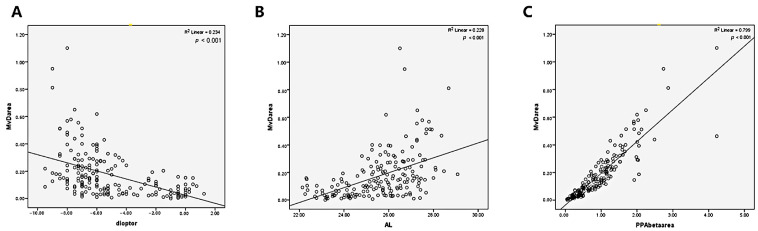
Correlation analysis between MvD area and spherical equivalent (SE), axial length (AL), and the area of PPA-β zone. MvD area was negatively correlated with (**A**) SE whereas positively correlated with (**B**) AL and (**C**) the area of PPA-β zone.

**Table 1 jpm-13-00289-t001:** Baseline characteristics of the included participants.

Group	N (Eyes)	Female	Age	SE ^1^	AL ^2^
High myopia group	95	32	36.21 ± 14.28	−7.14 ± 0.927	26.65 ± 0.88
Mild to moderate myopia group	110	31	32.30 ± 12.84	−2.51 ± 2.10	24.36 ± 1.15
*p* value	-	0.873	0.517	0.000	0.012

^1^ SE: spherical equivalent; ^2^ AL: axial length.

**Table 2 jpm-13-00289-t002:** Characteristics of participants having MvD ^1^ in PPA ^2^-α zone.

Group	N	*n* (Eyes with MvD in PPA-α Zone)	MvD ^2^ Area (mm^2^)	PPA-α Area (mm^2^)	Mean Density of Choroidal Flow in PPA-α Zone
High myopia group	95	21 (22.1%)	0.068 ± 0.050	0.336 ± 0.188	428.89 ± 312.20
Mild-moderate myopia group	110	63 (57.3%)	0.089 ± 0.082	0.235 ± 122	404.88 ± 432.47
*p* value	-	0.000	0.210	0.012	0.159

^1^ MvD: microvascular dropout; ^2^ PPA: peripapillary atrophy.

## Data Availability

Data available on request due to privacy restrictions. The data presented in this study are available on request from the corresponding author. The data are not publicly available due to privacy restrictions.

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
