# Peer review of "Microvascular Alterations of Peripapillary Choriocapillaris in Young Adult High Myopia Detected by Optical Coherence Tomography Angiography"

_jpm, 2023, doi:10.3390/jpm13020289_

Round 1

Reviewer 1 Report

This is an interesting study evaluating the peripapillary beta zone and its micorvasculature changes in lower grade and compare it with higher grade myopic patients.

One important ambiguity in this research is that how the MvD in PPA-beta zone is a representative of choriocapillaris vascular changes?

in discussion, what the authors really talked about their own work apart from limitation, is the first paragraph. however, some statements are not exact, such as "Here, we found MvDs can be detected in almost all PPA-β zones, but also in some PPA-α zones, which indicated microcirculatory changes of choroid prior to RPE and choroidal dystrophy"

I think by "choroidal dystrophy" they meant "choroidal atrophy"

In the above sentence, the fact that first and second event cannot be concluded from their work.

the same comment is applied for the sentence after the above mentioned sentence in the first paragraph of discussion.

the third paragraph of discussion is not really related to what they have done.

Author Response

We feel great thanks for your professional review work on our article. As you mentioned, there are several concerns that need to be addressed. According to your thoughtful suggestions, we have made extensive revisions which are listed below and marked blue in the corresponding context.

Q(1) One important ambiguity in this research is that how the MvD in PPA-beta zone is a representative of choriocapillaris vascular changes?

Response: Thank you for your valuable suggestions. There were publications that can be referred and we want to explain how the MvD served as a biomarker of choroidal vascular changes in high myopia as below (marked blue from line 57 to 71 and from line 281 to 286).

The choriocapillaris, restricted to the innermost part of the choroid, is composed of highly compacted capillaries separated by many tiny inter-capillary spaces that may grow with age and disease[11, 12]. Such inter-capillary spaces were strikingly captured by optical coherence tomography angiography (OCTA) and represented in the form of small dark regions, known as flow voids or microvascular dropouts (MvD)[13]. Lee et al.[14] confirmed that the MvDs found by OCTA were coincident with perfusion defects of indocyanine green angiography in glaucoma. Though there have been studies that focused on central macular choroidal features in eyes with high myopia[15-17], less attention has been paid to the peripapillary alterations by OCTA. This may result from limited avenues to represent peripapillary blood state as the image quality is easily affected by the distortion of the optic disc, a symptom usually seen in high myopes, thus creating unfavorable errors and reaching misleading conclusions. MvD, a newly-proposed concept that had never been investigated in high myopic populations, may bring vigor to improved precision in evaluating choroidal perfusion defects.

References

  1. R. F. Spaide. Choriocapillaris Flow Features Follow a Power Law Distribution: Implications for Characterization and Mechanisms of Disease Progression. Am J Ophthalmol 2016, 170:58-67. 10.1016/j.ajo.2016.07.023
  2. M. Al-Sheikh, N. Phasukkijwatana, et al. Quantitative OCT Angiography of the Retinal Microvasculature and the Choriocapillaris in Myopic Eyes. Invest Ophthalmol Vis Sci 2017, 58(4):2063-2069. 10.1167/iovs.16-21289
  3. E. Borrelli, D. Sarraf, et al. OCT angiography and evaluation of the choroid and choroidal vascular disorders. Prog Retin Eye Res 2018, 67:30-55. 10.1016/j.preteyeres.2018.07.002
  4. E. J. Lee, K. M. Lee, et al. Parapapillary Choroidal Microvasculature Dropout in Glaucoma: A Comparison between Optical Coherence Tomography Angiography and Indocyanine Green Angiography. Ophthalmology 2017, 124(8):1209-1217. 10.1016/j.ophtha.2017.03.039
  5. K. Sayanagi, Y. Ikuno, et al. Features of the choriocapillaris in myopic maculopathy identified by optical coherence tomography angiography. Br J Ophthalmol 2017, 101(11):1524-1529. 10.1136/bjophthalmol-2016-309628
  6. J. Ye, M. Wang, et al. Deep Retinal Capillary Plexus Decreasing Correlated With the Outer Retinal Layer Alteration and Visual Acuity Impairment in Pathological Myopia. Invest Ophthalmol Vis Sci 2020, 61(4):45. 10.1167/iovs.61.4.45
  7. S. S. Y. Lee, G. Lingham, et al. Choroidal Thickness in Young Adults and its Association with Visual Acuity. Am J Ophthalmol 2020, 214:40-51. 10.1016/j.ajo.2020.02.012

With the development of OCTA, choroidal MvD has frequently been reported in glaucoma cases as an indicator of vascular changes[35, 36]. Numerous studies have proved its relation to structural and functional alterations[37-41]. What is well known is that glaucomatous eyes exhibit a higher prevalence of myopia and myopia is one of the risk factors that induce primary open-angle glaucoma[42], based on which we trans-planted the concept of MvD in the context of myopia.

References

  1. H. L. Rao, S. Sreenivasaiah, et al. Choroidal Microvascular Dropout in Primary Angle Closure Glaucoma. Am J Ophthalmol 2019, 199:184-192. 10.1016/j.ajo.2018.11.021
  2. Y. H. Jo, K. R. Sung, et al. Comparison of Peripapillary Choroidal Microvasculature Dropout in Primary Open-angle, Primary Angle-closure, and Pseudoexfoliation Glaucoma. J Glaucoma 2020, 29(12):1152-1157. 10.1097/ijg.0000000000001650
  3. E. J. Lee, S. H. Lee, et al. Parapapillary Deep-Layer Microvasculature Dropout in Glaucoma: Topographic Association With Glaucomatous Damage. Invest Ophthalmol Vis Sci 2017, 58(7):3004-3010. 10.1167/iovs.17-21918
  4. H. L. Rao, S. Sreenivasaiah, et al. Choroidal Microvascular Dropout in Primary Open-angle Glaucoma Eyes With Disc Hemorrhage. J Glaucoma 2019, 28(3):181-187. 10.1097/ijg.0000000000001173
  5. J. A. Kim, D. H. Son, et al. Intereye Comparison of the Characteristics of the Peripapillary Choroid in Patients with Unilateral Normal-Tension Glaucoma. Ophthalmol Glaucoma 2021, 4(5):512-521. 10.1016/j.ogla.2021.02.003
  6. J. A. Kim, E. J. Lee, et al. Evaluation of Parapapillary Choroidal Microvasculature Dropout and Progressive Retinal Nerve Fiber Layer Thinning in Patients With Glaucoma. JAMA Ophthalmol 2019, 137(7):810-816. 10.1001/jamaophthalmol.2019.1212
  7. S. Lin, H. Cheng, et al. Parapapillary Choroidal Microvasculature Dropout Is Associated With the Decrease in Retinal Nerve Fiber Layer Thickness: A Prospective Study. Invest Ophthalmol Vis Sci 2019, 60(2):838-842. 10.1167/iovs.18-26115
  8. M. W. Marcus, M. M. de Vries, et al. Myopia as a risk factor for open-angle glaucoma: a systematic review and meta-analysis. Ophthalmology 2011, 118(10):1989-1994.e2. 10.1016/j.ophtha.2011.03.012

Q(2) In discussion, what the authors really talked about their own work apart from limitation, is the first paragraph. however, some statements are not exact, such as "Here, we found MvDs can be detected in almost all PPA-β zones, but also in some PPA-α zones, which indicated microcirculatory changes of choroid prior to RPE and choroidal dystrophy"

I think by "choroidal dystrophy" they meant "choroidal atrophy"

In the above sentence, the fact that first and second event cannot be concluded from their work.

the same comment is applied for the sentence after the above mentioned sentence in the first paragraph of discussion.

Response: Thank you for your valuable suggestions to improve our conclusions more logical and clearer. Our changes are shown below and marked blue from line 294 to 322.

Notably, the association between the microstructure of PPA-β zone and retinal vessel density has been investigated by Sung et al.[7]. The width of PPA zone was negatively correlated with superficial and deep parapapillary vessel density. At choroidal level, Hu et al.[43] acquired similar outcomes. Our results, from another perspective of MvD, confirmed the association between choroidal circulation and retinal structure. MvD was more frequently detected in the PPA-β zone and relevant to the PPA-β area, age, and AL, reassuring that the drop of choroidal perfusion is linked to retinal structural deterioration and myopia progression. Choroid situating between the sclera and BM is the destination of 70% of all the ocular blood flow for its highly-compacted capillary network[44]. Being provided oxygen and nourishment by choroidal circulation[45], the inner segments in the outer retina of the eye are able to support the photoreceptors, the highest rate of oxygen use per unit weight of tissue in the body[46], to sustain normal morphology and function. The choroid serves additional roles as a heat sink[44] and absorbing stray light, involving in immune response, and host defense[47], and has an influence in emmetropization[45]. It makes sense that during the elongation of progressively myopic eyes, the choroid is stretched and the vessels are impaired or even destroyed. Abnormalities of the retina state may be induced following the reduction of choroidal blood supply.

PPA-α is histologically defined by the presence of BM and an irregularly arranged RPE, which in some eyes appears to be rolled up at its end while PPA-β is characterized by the presence of BM and absence of RPE[23, 25]. Corresponding to the histological anatomy, the PPA-β zone represents an absolute scotoma in perimetry, and PPA-α zone a relative scotoma[48-50]. Therefore, PPA-β is the more severe form of retinal structural destruction developed from PPA-α. Here, we found MvDs can be detected in almost all PPA-β zones, but also in some PPA-α zones, which indicated the presence of microcirculatory defects of choroid before retinal degeneration from PPA-α into PPA-β. In another word, the changes in blood flow at the choroid level may be prior to RPE and choroidal atrophy. Such observations are explainable as insufficiency of life-sustaining materials originating from choroidal ischemia would be an integral part of apoptosis of RPE cells and consequent PPA development.

References

  1. M. S. Sung, H. Heo, et al. Microstructure of Parapapillary Atrophy Is Associated With Parapapillary Microvasculature in Myopic Eyes. Am J Ophthalmol 2018, 192:157-168. 10.1016/j.ajo.2018.05.022
  2. Y. X. Wang, R. Jiang, et al. Acute Peripapillary Retinal Pigment Epithelium Changes Associated with Acute Intraocular Pressure Elevation. Ophthalmology 2015, 122(10):2022-8. 10.1016/j.ophtha.2015.06.005
  3. J. B. Jonas, X. N. Nguyen, et al. Parapapillary chorioretinal atrophy in normal and glaucoma eyes. I. Morphometric data. Invest Ophthalmol Vis Sci 1989, 30(5):908-18.
  4. X. Hu, K. Shang, et al. Clinical features of microvasculature in subzones of parapapillary atrophy in myopic eyes: an OCT-angiography study. Eye (Lond) 2021, 35(2):455-463. 10.1038/s41433-020-0872-6
  5. L. M. Parver, C. Auker, et al. Choroidal blood flow as a heat dissipating mechanism in the macula. Am J Ophthalmol 1980, 89(5):641-6. 10.1016/0002-9394(80)90280-9
  6. D. L. Nickla and J. Wallman. The multifunctional choroid. Prog Retin Eye Res 2010, 29(2):144-68. 10.1016/j.preteyeres.2009.12.002
  7. N. D. Wangsa-Wirawan and R. A. Linsenmeier. Retinal oxygen: fundamental and clinical aspects. Arch Ophthalmol 2003, 121(4):547-57. 10.1001/archopht.121.4.547
  8. X. Yuan, X. Gu, et al. Quantitative proteomics: comparison of the macular Bruch membrane/choroid complex from age-related macular degeneration and normal eyes. Mol Cell Proteomics 2010, 9(6):1031-46. 10.1074/mcp.M900523-MCP200
  9. F. Rensch and J. B. Jonas. Direct microperimetry of alpha zone and beta zone parapapillary atrophy. Br J Ophthalmol 2008, 92(12):1617-9. 10.1136/bjo.2008.139030
  10. J. H. Meyer, M. Guhlmann, et al. Blind spot size depends on the optic disc topography: a study using SLO controlled scotometry and the Heidelberg retina tomograph. Br J Ophthalmol 1997, 81(5):355-9. 10.1136/bjo.81.5.355
  11. J. B. Jonas, G. C. Gusek, et al. Correlation of the blind spot size to the area of the optic disk and parapapillary atrophy. Am J Ophthalmol 1991, 111(5):559-65. 10.1016/s0002-9394(14)73698-0

Q(3) the third paragraph of discussion is not really related to what they have done.

Response: Thank you for your valuable suggestions. We intended to discuss the association between structural and vascular changes during the progression of myopia based on our observation of larger PPA-β zone and lower choroidal vessel density in the high myopia group (Figure 2). According to your helpful suggestions, we reorganized the context which is shown below and marked blue from line 256 to 281 in the revised version.

Our results showed that the high myopia group exhibited lower density of choroidal blood flow and larger PPA-β area (Figure 2A&B), indicating aggravating vascular and structural perturbations promote myopia progression. These findings are in agreement with previous studies that demonstrated less choroidal blood perfusion in high myopia using pulse amplitude and other relevant parameters[29, 30]. In parallel, decreased retinal nerve fiber layer and choroidal thickness accompanied by subsequent disc hemorrhage have been linked to myopia[31]. However, whether structural or vascular alterations change first or both change simultaneously has troubled experts and researchers for a long time. On the one hand, a decline in choriocapillary perfusion may arise from reduced blood demands secondary to choroidal degeneration occurring in highly myopic eyes. Jing Zhao et al.[32] found that axial elongation-associated choroidal thinning affected Haller's and Sattler's layers more markedly than the small-vessel layer. Lack of blood supply from media- and large-sized choroidal layer may also contribute to lower choriocapillary perfusion. Besides, it is also speculated that the progression of optic disc tilt during global elongation may directly and mechanically deteriorate peripapillary microvasculature[33]. For another hand, hypoxia has been regarded as essential for scleral extracellular remodeling during myopia development[34]. Visual perception may induce cell signaling pathway cascades that affect choroidal blood perfusion and scleral oxygenation, triggering a series of downstream events that diminish extracellular integrity and exacerbate scleral extension. This is a chicken-and-egg question in the light of which comes first, which warrants further prospective studies to determine the sequence between structural and vascular alterations.

References

  1. Y. F. Shih, I. H. Horng, et al. Ocular pulse amplitude in myopia. J Ocul Pharmacol 1991, 7(1):83-7. 10.1089/jop.1991.7.83
  2. Y. S. Yang and J. W. Koh. Choroidal Blood Flow Change in Eyes with High Myopia. Korean J Ophthalmol 2015, 29(5):309-14. 10.3341/kjo.2015.29.5.309
  3. C. Y. Kim, E. J. Lee, et al. Progressive retinal nerve fibre layer thinning and choroidal microvasculature dropout at the location of disc haemorrhage in glaucoma. Br J Ophthalmol 2021, 105(5):674-680. 10.1136/bjophthalmol-2020-316169
  4. J. Zhao, Y. X. Wang, et al. Macular Choroidal Small-Vessel Layer, Sattler's Layer and Haller's Layer Thicknesses: The Beijing Eye Study. Sci Rep 2018, 8(1):4411. 10.1038/s41598-018-22745-4
  5. J. He, Q. Chen, et al. Association between retinal microvasculature and optic disc alterations in high myopia. Eye (Lond) 2019, 33(9):1494-1503. 10.1038/s41433-019-0438-7

Reviewer 2 Report

The main topic addressed by this research is to detect main microvascular changes in peripapillary networks in young adults with high myopia by OCTA. In my opinion, the topic is scientifically relevant and interesting for the readers. The text is well-written. The methodology is accurate. Conclusions are consistent with the main topic. References are appropriate. Figures are presented in correct form and in good quality.

Author Response

Comments: The main topic addressed by this research is to detect main microvascular changes in peripapillary networks in young adults with high myopia by OCTA. In my opinion, the topic is scientifically relevant and interesting for the readers. The text is well-written. The methodology is accurate. Conclusions are consistent with the main topic. References are appropriate. Figures are presented in correct form and in good quality.

Response: We appreciate your positive review comments on our article. We have also revised the manuscript accordingly and marked blue in the corresponding text.

Reviewer 3 Report

The paper examines peripapillary microcirculation damage in myopic eyes. Myopia, especially high myopia, is a very important health problem that can be complicated by serious changes on the eye fundus. Microcirculation at the level of the peripapillary choriocapillaris is very important for the functioning and preservation of retinal nerve fibers, which leave the eyeball via the optic nerve head and conduct the impuls through the optic nerve and further into the bigger part of the visual path.

The results in the study are important because they show well what are the possible causes of peripapillary microcirculation damage (microvascular dropout) in myopic eyes, and with which factors are correlated.

Author Response

Comments: The paper examines peripapillary microcirculation damage in myopic eyes. Myopia, especially high myopia, is a very important health problem that can be complicated by serious changes on the eye fundus. Microcirculation at the level of the peripapillary choriocapillaris is very important for the functioning and preservation of retinal nerve fibers, which leave the eyeball via the optic nerve head and conduct the impuls through the optic nerve and further into the bigger part of the visual path.

The results in the study are important because they show well what are the possible causes of peripapillary microcirculation damage (microvascular dropout) in myopic eyes, and with which factors are correlated.

Response: We appreciate your positive review comments on our article. We have also revised the manuscript accordingly and marked blue in the corresponding text.

Reviewer 4 Report

I want to congratulate the authors on an excellent article. One note is that it would have been better to use a swept source oct device

Author Response

Q: I want to congratulate the authors on an excellent article. One note is that it would have been better to use a swept source oct device.

Response: We feel great thanks for your positive review comments on our article. According to your professional suggestions, we have made a supplement on discussing the possibility of a swept source OCT device from line 240 to 243. The changes are shown below and marked blue in the corresponding paragraphs.

Recently, the emergence of the newest generation of swept-source OCT (SS-OCT) with a longer wavelength (1050~1060 nm) made it possible for the light to penetrate RPE and image clearer choroidal vasculature. It is apt to adopt SS-OCT in future studies for higher imaging speed and better signal-to-noise ratio.